# Calculation of Permeability Coefficients from Solute Equilibration Dynamics: An Assessment of Various Methods

**DOI:** 10.3390/membranes12030254

**Published:** 2022-02-23

**Authors:** Margarida M. Cordeiro, Armindo Salvador, Maria João Moreno

**Affiliations:** 1Coimbra Chemistry Centre-Institute of Molecular Sciences (CQC-IMS), University of Coimbra, 3004-535 Coimbra, Portugal; mmc.margarida0@gmail.com; 2Department of Chemistry, University of Coimbra, 3004-535 Coimbra, Portugal; 3CNC—Centre for Neuroscience Cell Biology, University of Coimbra, UC-Biotech, Parque Tecnológico de Cantanhede, Núcleo 04, Lote 8, 3060-197 Cantanhede, Portugal; 4Institute for Interdisciplinary Research, University of Coimbra, Casa Costa Alemão, 3030-789 Coimbra, Portugal

**Keywords:** membrane permeation, kinetic modelling, lipophilicity, permeation of weak acids, membrane sequestration, drug availability, lipid membranes, liposomes

## Abstract

Predicting the rate at which substances permeate membrane barriers in vivo is crucial for drug development. Permeability coefficients obtained from in vitro studies are valuable for this goal. These are normally determined by following the dynamics of solute equilibration between two membrane-separated compartments. However, the correct calculation of permeability coefficients from such data is not always straightforward. To address these problems, here we develop a kinetic model for solute permeation through lipid membrane barriers that includes the two membrane leaflets as compartments in a four-compartment model. Accounting for solute association with the membrane allows assessing various methods in a wide variety of conditions. The results showed that the often-used expression *P*_app_ = *β* × *r*/3 is inapplicable to very large or very small vesicles, to moderately or highly lipophilic solutes, or when the development of a significant pH gradient opposes the solute’s flux. We establish useful relationships that overcome these limitations and allow predicting permeability in compartmentalised in vitro or in vivo systems with specific properties. Finally, from the parameters for the interaction of the solute with the membrane barrier, we defined an intrinsic permeability coefficient that facilitates quantitative comparisons between solutes.

## 1. Introduction

To reach their target sites from the site of administration, drugs must cross a series of biological membranes. Insufficient permeability through any of these membranes is a major cause of attrition in drug development. For this reason, much effort has been devoted to characterise the rate of permeation of large sets of drug-like molecules, seeking to achieve predictive power. Multiple experimental methodologies have been developed for this purpose, using either cell monolayer membranes, [1,2,3,4] or simpler lipid membranes [5,6,7,8,9]. The former methodologies have the advantage of providing a more direct estimate of the behaviour in vivo, as they account not only for passive permeation of the lipid bilayers but also for processes such as active transport, efflux by pumps or sequestration in cells. However, to gain predictive power, it is necessary to understand the rules that govern each permeation pathway. For this, simpler model systems such as lipid membranes are required.

Among the methods available to characterise the rate of permeation of drug-like molecules through lipid membranes, the pH variation assay is particularly relevant [10,11,12,13,14]. This is because it allows following the permeation of weak acids or bases, which most drugs are, even when they do not possess fluorescent groups. This versatility prompts the characterisation of the passive permeation of a large number of structurally unrelated drug-like molecules, thus yielding a wealth of data that will allow (machine- and otherwise) learning the general rules that govern this process. This methodology uses lipid vesicles as mimetic systems of biological membranes. The permeability coefficient of the solute is calculated from the dynamics of its entry into the vesicles, which is accompanied by a pH variation. The latter is reported by an encapsulated pH-sensitive fluorescent probe. In spite of the high potential of this approach, the permeability coefficients obtained are not always consistent with those from other approaches [7,8,15,16,17]. Moreover, its validity has been questioned, mostly regarding the possible contribution of the rate of acid-base equilibria [18,19], although it was concluded by the same authors that this could not be the rate-limiting step [18]. Here, we assess different approaches for the calculation of permeability coefficients from solute equilibration dynamics, and evaluate whether the mis-application of some of these approaches contributes to the lack of quantitative agreement with different experimental methodologies.

The dynamics of permeation through the barrier may be evaluated from the initial rate of permeation (less than 10% of the total variation), or from the characteristic constant (*β*, a list of abbreviations is presented in the Appendix A) at which the equilibrium is approached. The amount of solute in the acceptor compartment ($nS_{A}$) is usually zero at *t* = 0; its increase over time is often described by a mono-exponential function, Equation (1). In this case, the initial rate is related with the characteristic constant by Equation (2).
(1)$$\begin{array}{c} nS_{A}\;\left(t\right)\;=\;nS_{A}\;\left(\infty \right)\;+\;\left[nS_{A}\;\left(0\right)\;-\;nS_{A}\;\left(\infty \right)\right]\;e^{-\beta t}\\ \;\;\;\;\;=\;nS_{A}\;\left(\infty \right)\left(1\;-\;e^{-\beta t}\right)\;\mathrm{when}\;nS_{A}\;\left(0\right)\;=0 \end{array}$$
(2)$${\frac{dnS_{A}}{dt}|}_{0}=nS_{A}\left(\infty \right)\;\beta$$

For a given intrinsic permeability through the barrier, the characteristic constant (β) at which the equilibrium between the two compartments is attained depends on the system topology. Namely, on the compartments’ relative volumes, and on the surface area separating them. In contrast, the permeability coefficient ($P_{\mathrm{app}}$) is not dependent on the system geometry, and is defined by:(3)$$P_{\mathrm{app}}=\frac{{\frac{dnS_{A}}{dt}|}_{0}}{{\left[S_{D}\right]}_{0}A}=\frac{{\frac{dnS_{A}}{dt}|}_{0}}{nS_{D}\left(0\right)}\frac{V_{D}}{A}.$$

The derivation of the above equation assumes that the diffusion of the permeating molecule within the donor and acceptor compartments is faster than permeation through the barrier (non-diffusion-controlled interaction with the barrier), and that there is no significant accumulation of the permeating molecule in the barrier.

Combining Equations (2) and (3), allows the calculation of $P_{\mathrm{app}}$ from the observed characteristic constant of equilibration between the donor and acceptor compartments:(4)$$P_{\mathrm{app}}=\beta \frac{nS_{A}\left(\infty \right)\;}{nS_{D}\left(0\right)}\frac{V_{D}}{A}$$

When the efflux of very polar molecules from lipid vesicles is considered, Equation (4) may be greatly simplified because the volume of the aqueous medium inside the vesicles (the donor, $V_{D}$) is usually much smaller than the volume outside the vesicles (the acceptor, $V_{A}$), and because the permeating polar molecule does not strongly associate with the lipid membrane. In these conditions, it may be assumed that all solute permeates the membrane. That is, the amount of solute in the acceptor compartment at equilibrium equals the initial amount in the donor compartment, leading to
(5)$$P_{\mathrm{app}}=\beta \frac{V_{D}}{A}$$

If the vesicles are spherical and monodisperse with radius r, Equation (5) further simplifies to
(6)$$P_{\mathrm{app}}^{r}=\beta \frac{r}{3}$$

This equation has been used when the efflux of the solute is being followed [20,21], in the case of influx of solute into the aqueous compartment of the lipid vesicles [19,22,23,24,25,26], and for distinct solutes permeating in both directions [27,28,29,30,31]. Some aspects regarding the validity of this equation have been discussed. However, the domain of validity of this approach has not been adequately explored. Given the small relative volume of the acceptor compartment in influx assays, and since the characteristic constant for equilibration decreases (slows down) as the amount of solute that needs to permeate to achieve equilibrium increases [6], we anticipate that $P_{\mathrm{app}}^{r}$ is a poor estimate of $P_{\mathrm{app}}$ in some situations.

Another important concern is the validity of the assumption of negligible accumulation in the membrane. While this may be a good approximation for very polar solutes, drug-like molecules are usually somewhat lipophilic and may accumulate significantly in the membrane.

In this work we develop a kinetic model that allows the description of the dynamics of solute equilibration between two aqueous compartments separated by a lipid membrane. In contrast to previous models, the actual thickness of the lipid bilayer is accounted for, leading to four compartments. Namely, the aqueous media outside and inside the vesicles, and the outer and inner membrane leaflets. The model is developed for the permeation of non-ionisable solutes and weak acids, in the latter case including their ionisation equilibria in both the aqueous compartments and when associated with the membrane. The explicit inclusion of the lipid membrane allows for the consideration of the intrinsic parameters for its interaction with the solute. Namely, the partition coefficient between the aqueous media and the adjacent membrane leaflet, and the rate constant for translocation between the two membrane leaflets. By changing the geometry of the system while maintaining the intrinsic solute parameters, one may evaluate the validity of the different methods to calculate the solute’s permeability coefficient. Throughout the manuscript, several case-specific equations are derived that may be used to calculate the solute’s permeability coefficient from the equilibrium dynamics of its influx into the vesicles. Relationships to calculate the intrinsic permeability coefficient of the solute from its parameters for interaction with the lipid membranes are also provided, both for non-ionisable solutes and weak acids. The adaptation of the model to weak bases is straightforward and is provided in the Appendix A.

## 2. Materials and Methods

Two kinetic models were developed. The first model describes the permeation of molecules that are globally neutral and do not change their ionisation state. The second one assumes that only the neutral form of a weak acid can permeate the membrane. The latter model considers all the acid-base equilibria, both in the water phases and when the solute is associated with both membrane leaflets. This section provides a detailed description of the geometric considerations and the kinetic models.

### 2.1. Geometric Parameters

We consider that the lipid is distributed in *N* spherical lipid vesicles ($N_{\mathrm{vesicles}}$) with outer radius $r_{o}$ and lipid bilayer thickness h in a suspension with a total volume $V_{T}$. Therefore, the volumes of the outer aqueous medium ($V_{\mathrm{wo}}$) and lipid membrane leaflet ($V_{\mathrm{lo}}$), and inner lipid membrane leaflet ($V_{\mathrm{li}}$) and aqueous medium ($V_{\mathrm{wi}}$) are given by Equations (7) to (10), respectively:(7)$$V_{\mathrm{wo}}=V_{T}-\frac{4}{3}\pi \;r_{o}^{3}N_{\mathrm{vesicles}},$$
(8)$$V_{\mathrm{lo}}=\frac{4}{3}\pi \left[r_{o}^{3}-{\left(r_{o}-\frac{h}{2}\right)}^{3}\right]N_{\mathrm{vesicles}},$$
(9)$$V_{\mathrm{li}}=\frac{4}{3}\pi \left[{\left(r_{o}-\frac{h}{2}\right)}^{3}-{\left(r_{o}-h\right)}^{3}\right]N_{\mathrm{vesicles}},$$
(10)$$V_{\mathrm{wi}}=\frac{4}{3}\pi {\left(r_{o}-h\right)}^{3}N_{\mathrm{vesicles}}.$$

The number of vesicles can be computed from the lipid concentration in the system ($c_{L}$) and the area per lipid molecule ($a_{L}$) through equation:(11)$$N_{\mathrm{vesicles}}=\frac{c_{L}\;V_{T}\;N_{A}}{N_{L\;per\;\mathrm{vesicle}}}\;;\;N_{L\;per\;\mathrm{vesicle}}=\frac{4\pi \left[r_{o}^{2}+{\left(r_{o}-h\right)}^{2}\right]}{a_{L}},$$
where $N_{A}$ is Avogadro’s number. The area of the surface separating the two membrane leaflets was obtained through equation:(12)$$A_{\mathrm{io}}=4\pi {\left(r_{o}-\frac{h}{2}\right)}^{2}N_{\mathrm{vesicles}}.$$

The bilayer thickness is estimated as explained in Appendix B.

### 2.2. Model Description

The model for permeation of non-ionisable solutes (Model I) describes the permeation process as a three-step mechanism: insertion in the outer monolayer, translocation between the outer and inner leaflet, and desorption from the bilayer (Figure 1A,B). It is assumed that the insertion and desorption processes equilibrate within the time scale of the translocation step, as the latter is the single rate-limiting step for most solutes.

The dynamics of this system is defined by the differential Equation (13) for the aggregated slow variable $nS_{i}^{}$, which is the amount of solute in the inner compartments (including the inner membrane leaflet and the inner aqueous medium, $nS_{\mathrm{li}}$ and $nS_{\mathrm{wi}}$, respectively). The parameters $k_{\mathrm{lio}}^{S}$ and $k_{\mathrm{loi}}^{S}$ correspond to the rate of solute flip-flop from the inner into the outer and from the outer into the inner membrane leaflets, respectively.
(13)$$\frac{d\;nS_{i}^{}\left(t\right)}{dt}=-k_{\mathrm{lio}}^{S}\;nS_{\mathrm{li}}\left(t\right)+k_{\mathrm{loi}}^{S}\;nS_{\mathrm{lo}}\left(t\right)$$

The differential equation expressed in terms of the local concentration of solute is obtained by dividing each variable by the volume of the corresponding compartment $(V_{i}=V_{\mathrm{wi}}+V_{\mathrm{li}}$ in the case of the aggregated variable $S_{i}^{})$, and is given by:(14)$$\frac{dS_{i}^{}\left(t\right)}{dt}=-k_{\mathrm{lio}}^{S}\;S_{\mathrm{li}}\left(t\right)\frac{V_{\mathrm{li}}}{V_{i}}+k_{\mathrm{loi}}^{S}\;S_{\mathrm{lo}}\left(t\right)\frac{V_{\mathrm{lo}}}{V_{i}}.$$

The local concentration of all the species ($S_{\mathrm{wo}},\;S_{\mathrm{wi}},\;S_{\mathrm{lo}}\;\mathrm{and}\;S_{\mathrm{li}}$) is obtained from the conservation and equilibrium relationships between them (see Appendix C for details).

In the model for the permeation of weak acids (Model II, Figure 1C), it is assumed that only the uncharged (i.e., protonated, $SH_{\mathrm{wo}},\;SH_{\mathrm{wi}},\;SH_{\mathrm{lo}}\;\mathrm{and}\;SH_{\mathrm{li}}$) species permeate. This model includes a pH-buffer in the aqueous phase outside the vesicles (protonated and deprotonated forms $BH_{\mathrm{wo}}$ and $BD_{\mathrm{wo}}$, respectively), and a pH-sensitive probe inside the vesicles $\;(PH_{\mathrm{wi}}$ and $PD_{\mathrm{wi}})$ that reports the variation in the pH due to the solute’s permeation. The protonation and deprotonation equilibria for the buffer, probe, and solute are considered, as well as the partition of the solute species between the aqueous compartments and the adjacent membrane leaflets, and the translocation of the protonated solute between leaflets.

The presence of the buffer in the external aqueous medium maintains the pH in the outer compartments unchanged. In the aqueous phase, the protonated solute can deprotonate ($SD_{\mathrm{wo}}$), and both solute species can associate with the membrane’s outer leaflet. Once there, the solute can change its protonation state but only the neutral form is allowed to flip-flop across the hydrophobic core of the membrane, with the rate constants $k_{\mathrm{loi}}^{\mathrm{SH}}$ and $k_{\mathrm{lio}}^{\mathrm{SH}}$ for outer → inner and inner → outer flip-flops, respectively.

When the protonated solute is located near the surface of the inner membrane leaflet, it can release a proton into the internal aqueous medium, generating the conjugated base. Both the weak acid and its conjugated base may move from the inner membrane leaflet into the inner aqueous phase, where they equilibrate with each other. The protons released equilibrate with the fluorescent probe. As the protonated solute permeates into the inner compartments and equilibrates with its deprotonated form, the pH in the inner aqueous compartment decreases. The ensuing change in the fraction of the fluorescent probe species allows following the permeation of the weak acid.

In this model, the quasi-equilibrium approximation is applied to all the protonation and deprotonation steps, as those processes are usually quite fast, both in bulk water and at the lipid bilayer surface [32]. Partition processes are also considered in quasi-equilibrium. That is, it is assumed that the solute’s insertion into and desorption from the membrane are faster than its flip-flop between the membrane leaflets. This approximation is expected to apply to solutes with low-to-moderate lipophilicity, although it may break down for very lipophilic solutes [6,33,34,35,36]. With those approximations, the permeation of the weak acid is also described by a single differential equation expressed in terms of the aggregated variable $S_{i}^{}$, which represents the total solute that has crossed the barrier (i.e., the sum of $SH_{\mathrm{wi}},\;SD_{\mathrm{wi}},\;SH_{\mathrm{li}}$ and $SD_{\mathrm{li}}$),
(15)$$\frac{dS_{i}^{}\left(t\right)}{dt}=-k_{\mathrm{lio}}^{\mathrm{SH}}\;SH_{\mathrm{li}}\left(t\right)\frac{V_{\mathrm{li}}}{V_{i}}+k_{\mathrm{loi}}^{\mathrm{SH}}\;SH_{\mathrm{lo}}\left(t\right)\frac{V_{\mathrm{lo}}}{V_{i}}.$$

The rates of change of the aggregated variables $nS_{o}^{}$ (the total amount of solute in the outer compartments) and $nH_{i}^{}$ (the total amount of labile protons in the inner compartments), are related to that for $nS_{i}^{}$ by:(16)$$\begin{array}{l} \frac{d\;nS_{o}^{}\left(t\right)}{dt}=-\frac{d\;nS_{i}^{}\left(t\right)}{dt},\\ \frac{d\;nH_{i}^{}\left(t\right)}{dt}=\frac{d\;nS_{i}^{}\left(t\right)}{dt}. \end{array}$$

The local concentration of all species ($SH_{\mathrm{wo}}$, $SD_{\mathrm{wo}}$, $SH_{\mathrm{wi}}$, $SD_{\mathrm{wi}}$, $SH_{\mathrm{lo}}$, $SD_{\mathrm{lo}}$, $SH_{\mathrm{li}}$, $SD_{\mathrm{li}}$, $BH_{\mathrm{wo}}$, $BD_{\mathrm{wo}}$, $PH_{\mathrm{wi}}$, $PD_{\mathrm{wi}}$, $H_{\mathrm{wo}}$ and, $H_{\mathrm{wi}}$) were calculated from the conservation and equilibrium relationships (see Appendix C for details).

The numerical integration of the differential equations was carried out in Mathematica^TM^ v.12.2 [37]. The considered simulation parameters are shown in Table 1.

## 3. Results and Discussion

The detailed models above allow evaluating the relationship between the observed characteristic constant for solute equilibration, the intrinsic parameters for solute-membrane interaction (affinity and flip-flop rate constant), and the geometry of the compartmentalised system. They also permit assessing the validity of the equations commonly used to estimate solutes’ permeability coefficients. This is an important parameter that is independent of system topology, and may thus be used to calculate the rate of equilibration in any system that is compartmentalised by lipid membranes.

The first section discusses the case of non-ionisable solutes. This represents a simple situation, because their equilibration between the two aqueous media only depends on the concentration gradient. Two situations will be analysed, one for solutes that have equal affinity for the membrane and aqueous media (non-lipophilic solute) and the other for lipophilic solutes. The first situation allows evaluating the effect of system topology, while the later allows evaluating the effect of solute sequestration in the membrane.

The second section examines the case of solutes that can change their ionisation state (weak acids), with membrane permeation of the neutral species only. In this case, solute permeation and the subsequent ionisation equilibria in the acceptor compartment led to a pH gradient across the membrane. Because we assume that the membrane is impermeable to charged species (including protons) the pH gradient at equilibrium is counterbalanced by an opposite concentration gradient of the anionic solute. That is, net flow stops before the solute concentrations equalise between donor and acceptor compartments. The solute gradient will be stronger the stronger the generated pH gradient. The effect of solute sequestration in the membrane will be evaluated. For simplicity, the same ionisation equilibria for the solute in the aqueous media and associated with the membrane was considered, although for solutes deeply inserted in the membrane stabilisation of the neutral species when in the non-polar membrane compartment should occur [7,40,41,42,43,44].

### 3.1. Non Ionisable Solutes

The intrinsic rate of membrane translocation (flip-flop) considered for the non-ionisable solutes is similar to that obtained experimentally for the fluorescent amphiphile NBD-Cn, with a nitrobenzoxadiazole as the polar group and an alkyl non-polar tail [6,45,46]. The rate of flip-flop was not strongly dependent on the length of the alkyl chain, varying from 0.2 s^−1^ for n = 16, to 14 s^−1^ for n = 2. In these simulations we have considered the intermediate value of 1 s^−1^ for the rate of translocation of the neutral solute. However, the affinity for the membrane was strongly influenced by the length of the alkyl chain, as expected: the partition coefficient for a POPC bilayer was 5 × 10^4^ for n = 8, and 20 for the nitrobenzoxadiazole without alkyl chain (n = 0) [6,47]. The sections below consider solutes with different partition coefficients between the aqueous media and the membrane (1, 10^2^ and 10^4^) in order to evaluate the effect of membrane partition on the overall rate of solute permeation.

#### 3.1.1. Non-Lipophilic Solutes

The simulated time evolution of the solute concentration in the aqueous compartments inside the lipid vesicles is shown in Figure 2A. Equilibration is very fast for the smaller vesicles considered ($r_{o}$ = 25 nm), and becomes slower as the vesicle size increases. The solute equilibration is well described by a mono-exponential function; the characteristic constants are shown in Figure 2B as a function of the vesicle radius.

The initial rate of solute permeation was very similar in all situations, being equal to the intrinsic rate of crossing the membrane barrier ($k_{\mathrm{loi}}^{S}$ = 1 s^−1^) multiplied by the amount of solute molecules available to cross the barrier ($nS_{\mathrm{lo}}(0)≅4\times 10^{-4}\times nS_{T}$ for the conditions used in the simulations: $c_{L}^{}$ = 1 mM and $K_{P}$ = 1), Equation (17). As the vesicle radius increases, the volume of the inner aqueous compartment increases as well, and more solute needs to permeate the membrane for equilibrium to be achieved (Figure 2C). This leads to a strong dependence of the equilibration time on vesicle size.
(17)$${\frac{dnS_{A}}{dt}|}_{t=0}=nS_{lo}\;k_{\mathrm{loi}}^{S}=nS_{T}\frac{K_{P}\frac{V_{\mathrm{lo}}}{V_{D}}}{1+K_{P}\frac{V_{\mathrm{lo}}}{V_{D}}}\;k_{\mathrm{loi}}^{S}$$

In the equation above, $V_{D}\;\mathrm{and}\;V_{\mathrm{lo}}$ stand for the total volume of the donor compartment and that of the membrane in the donor compartment, respectively.

The permeability coefficient calculated from the general Equation (4), $P_{\mathrm{app}}^{}$, is 2 × 10^−8^ dm s^−1^ for all vesicle sizes considered. However, the $P_{\mathrm{app}}^{}$ calculated from the simplified Equation (6), $P_{\mathrm{app}}^{r}$, depends on the radius of the vesicles, being higher for very small and for very large vesicles (Figure 3). To understand this result, one must consider the assumptions that are needed to obtain the simplified equation from the general equation for permeation, Equation (4). In the simulations performed in this section, at equilibrium the local concentration of solute in the donor (D) and acceptor (A) compartments is the same:(18)$$\frac{nS_{A}\left(\infty \right)}{V_{A}}=\frac{nS_{D}\left(\infty \right)}{V_{D}}=\;\frac{\left(nS_{D}\left(0\right)-nS_{A}\left(\infty \right)\right)}{V_{D}}\;\;\Leftrightarrow \;\;\frac{nS_{A}\left(\infty \right)}{nS_{D}\left(0\right)}=\frac{V_{A}}{V_{T}}$$

Replacing this relationship in the general Equation (4) yields:(19)$$P_{\mathrm{app}}^{\mathrm{rV}}=\beta \frac{V_{A}V_{D}}{V_{T}}\;\frac{1}{A}=\beta \frac{r}{3}\left(1-\frac{V_{A}}{V_{T}}\right)$$

Thus, as the relative volume of the acceptor compartment increases, the simplified equation becomes less accurate in the prediction of the solute permeability coefficient. For vesicles with a radius up to 500 nm and a total lipid concentration up to 1 mM, the volume of the inner aqueous compartment accounts for less than 5 % of the total volume, and, thus, $P_{\mathrm{app}}^{r}$ is a good approximation. However, for larger vesicles, the aqueous volume inside the vesicles is not negligible and causes the simplified equation to overestimate the solute permeability coefficient.

The size dependence observed for $P_{\mathrm{app}}^{r}$ with very small vesicles ($r_{o}$ < 100 nm) is due to the significant contribution of the membrane to the total volume of the acceptor compartments, and is corrected only if the aqueous inner volume of the vesicles is considered (which is equivalent to consider the internal vesicle radius ($r_{i}$) in the calculation of the barrier surface). The value of $P_{\mathrm{app}}^{}$ also depends on vesicles size. It increases with decreasing vesicle size when using $r_{o}$ or $r_{\mathrm{io}}$ for the calculation of the barrier surface, and decreases when using $r_{i}$. This is because for vesicles smaller than 100 nm the membrane curvature is significant, and the volume of the inner leaflet is smaller than that of the outer. Thus, the local concentration of solute at equilibrium is the same, but the amount of solute in the inner leaflet is smaller than that in the outer leaflet. This is expected to lead to an apparent increase in the permeability coefficient, and it is in quantitative agreement with that observed for $P_{\mathrm{app}}^{r}$ when considering $r_{\mathrm{io}}$. In fact, the interleaflet surface is the relevant barrier surface, because the non-polar centre of the bilayer was the barrier for the permeation of polar solutes. Accordingly, $r_{\mathrm{io}}$ will be considered the relevant radius hereafter.

#### 3.1.2. Lipophilic Solutes

Most drugs associate efficiently with the lipid membranes, with the partition coefficient from water to a fluid membrane ($K_{P}$) being usually in the 10^2^ to 10^4^ range [16,17,48,49,50,51]. The results obtained for the amount of solute in the vesicles’ inner compartments $(nS_{i},$ membrane inner leaflet *plus* aqueous medium), the corresponding rate of equilibration, and the estimated permeability coefficient are shown in Figure 4 as a function of $K_{P}.$ The vesicle radius was fixed at 100 nm, which corresponds to the largest unilamelar vesicles that may be prepared by extrusion. The rate of equilibration (*β*) and the value of $P_{\mathrm{app}}^{}$ calculated from the general Equation (4) strongly increase with increasing affinity of the solute for the membrane. This was expected because the fraction of solute molecules that are in the vicinity of the permeability barrier ($nS_{\mathrm{lo}}/nS_{T}$) increased. In the extreme situation where all the solute molecules were associated with the membrane, the rate of equilibration becomes equal to the intrinsic rate of exchange between the two membrane leaflets $(k_{\mathrm{loi}}^{S}+k_{\mathrm{lio}}^{S}),$ which in the model was 2 s^−1^ if the membrane curvature was negligible. However, the increase in $P_{\mathrm{app}}^{}$ (or *β*) is not proportional to $K_{P}$, in contrast to the usually assumed relationship, the Meyer–Overton Equation (20):(20)$$P=K_{P}\frac{D}{h}$$

In this equation, *D* is the solute diffusion coefficient through the barrier and is related with the flip-flop rate constant considered in the current model, and *h* is the thickness of the barrier, both parameters being fixed in the simulations shown in Figure 4.

This non-linear effect of $K_{P}$ reflects the relationship between the fraction of solute molecules in the membrane outer leaflet at the beginning of permeation. For the 1 mM lipid concentration considered in these simulations, 10% of the solute in the outer compartment is associated with the membrane for $K_{P}$ = 3 × 10^2^. Lower partition coefficients lead to proportionally lower fractions of solute in the membrane; a linear decrease is observed for $P_{\mathrm{app}}^{}$. However, the effect of increasing $K_{P}$ (i.e., lipophilicity) gradually levels-off at 100% of the solute at the membrane. Thus, increasing $K_{P}$ 10-fold and 100-fold (to 3 × 10^3^ and 3 × 10^4^, respectively) increases the fraction of solute associated with the membrane by just 5-fold and an additional 2-fold, respectively. In the latter condition, 92 % of the solute in the outer compartments is associated with the membrane.

If the permeability coefficient is calculated using the simplified equation, the deviations from the expected behaviour are even more severe, with $P_{\mathrm{app}}^{r}$ and $P_{\mathrm{app}}^{\mathrm{rV}}$ increasing less than 30-fold when $K_{P}$ increases from 1 to 10^4^. This is because in the derivation of (Equation (6) and (19)) it is assumed that the solute concentration in the donor and acceptor compartments is equal at equilibrium, which is not valid due to the higher contribution of the membrane in the case of the acceptor compartment ($V_{\mathrm{li}}/V_{\mathrm{wi}}\gg V_{\mathrm{lo}}/V_{\mathrm{wo}}$). Therefore, the simplified equation cannot be used for lipophilic solutes, even when their affinity for the membrane is only moderate.

To further evaluate the adequacy of the proposed equations to calculate the permeability coefficient in the case of lipophilic solutes, the dependence of the dynamic parameters on the radius of the vesicles was simulated for the case of high (*K*_P_ = 10^4^) and moderate (*K*_P_ = 10^2^) solute lipophilicity, Figure 5 and Appendix A, respectively.

For the case of high affinity, the characteristic constant for equilibration is almost unaffected by the size of the vesicles, varying only from 1.99 s^−1^ to 1.77 s^−1^ when the radius increases from 25 to 5000 nm (Figure 5A,B). This is close to the limit attained when all solute is associated with the outer membrane leaflet at the beginning of the permeation. This limit is slightly higher than 2 for very small vesicles due to the high curvature and the corresponding higher relative volume of the outer membrane leaflet. In this case, the amount of solute that needs to permeate the barrier to achieve equilibrium is smaller, and therefore the equilibrium is attained faster.

The permeability coefficient calculated from the general equation slightly decreases with *r* for very large vesicles. To understand this effect, one should consider the limit situation of infinite solute lipophilicity, with all solute being associated with the membrane. In this case, in the absence of significant membrane curvature, the equilibration characteristic constant is equal to the limit value and independent of vesicle radius. The amount of solute in the acceptor compartment is also independent of *r*, being half the total amount of solute. However, when $P_{\mathrm{app}}^{}$ is calculated from the general equation, it decreases with the increase in *r*, because the volume of the donor compartment decreases. This reflects the inadequacy of the equation in the extreme situation of very lipophilic solutes, when the contribution of the aqueous medium is null or negligible. In this case, only the membrane should be considered, and its volume is independent of the vesicles’ radius. The inclusion of the aqueous compartment leads to a decrease in the volume of the donor compartment as *r* increases. As a consequence, there is an artificial increase in the concentration of solute in this compartment at *t* = 0, which leads to a decrease in the calculated permeability coefficient.

When the simplified equation is used to calculate $P_{\mathrm{app}}^{},$ the dependence with the vesicle radius is very strong, increasing by over two orders of magnitude as *r* increases from 25 to 5000 nm. This is only partially corrected when considering that the volume of the donor compartment differs from the total volume of the system, $P_{\mathrm{app}}^{\mathrm{rV}}$, showing that the strong dependence with *r* is not due to this approximation. Instead, it is the assumption that the fraction of solute in the acceptor compartment at equilibrium is equal to fractional volume occupied by the acceptor compartments (Equation (18)) that makes the simplified equation inadequate to describe the dependence of $P_{\mathrm{app}}^{r}$ with *r*. The increase in the vesicles’ size leads to an increase in volume of the acceptor aqueous compartment, but the total volume of membrane inner leaflets remains unchanged. Because most solute is associated with the membrane, the equilibrium amount of solute in the acceptor compartment is essentially unchanged.

Solutes with a moderate lipophilicity originate an intermediate situation (Appendix A), with a significant contribution from both aqueous and membrane compartments. In this case, the general equation yields an excellent description of the system, with the calculated $P_{\mathrm{app}}^{}$ being essentially independent of the system topology. On the other hand, the simplified equation leads to a strong dependence with the vesicle radius, with $P_{\mathrm{app}}^{r}$ and $P_{\mathrm{app}}^{\mathrm{rV}}$ varying by more than an order of magnitude.

### 3.2. Weak Acids

#### 3.2.1. Non-Lipophilic Solutes

In these simulations, it is assumed that only the neutral (protonated) form of the solute permeated the lipid membrane, with the same rate constant as considered above for non-ionisable solutes. However, both solute species (protonated and deprotonated) are assumed to interact with the membrane, in this case with equally low affinity, $K_{P}$ = 1. This models the situation where a polar solute associates with the polar portion of the membrane, with both neutral and charged species interacting with similar affinity.

The results obtained for the accumulation of solute in the inner compartments of vesicles with a radius between 25 and 500 nm are shown in Figure 6. Larger vesicles are not considered because, as shown for non-ionisable solutes, the assumption of a negligible contribution of the vesicles’ inner aqueous compartment breaks down, and because they are rarely used in permeation experiments. The value considered for the solute ionisation equilibrium is 10^−7^ ($pK_{a}$ = 7); the initial pH in both compartments is 7. The results are compared to those for a non-ionisable non-lipophilic solute to facilitate the interpretation of the effect of ionisation.

The rate of equilibration of the weak acid is slower than for the case of a non-ionisable solute. This is because only the neutral species permeated the membrane, and this species accounts for only half of the solute in the outer membrane leaflet. This effect was expected to lead to a 50% decrease of *β*. However, *β* decreases only to ~0.8 of the value observed for the non-ionisable solute. This discrepancy occurs because less solute permeates the membrane into the inner compartments, and therefore the equilibrium is attained earlier. The lower amount of solute in the acceptor compartments ensues from the decrease in the pH in the acceptor compartments, due to the protons released as the inflowing weak acid equilibrates with its conjugated base. This decrease in the pH shifts the acid/base equilibrium towards the acid form. Moreover, at equal concentrations of the acid form in both donor and acceptor compartments, the concentration of the conjugated base is lower in the acceptor than in the donor compartment (Figure 7A).

The calculated $P_{\mathrm{app}}^{r}$ and $P_{\mathrm{app}}^{}$ are shown in Figure 6C. The general equation leads to a permeability coefficient equal to half of that observed for the non-ionisable solute, in agreement with the fact that only half of the solute is able to permeate the membrane. However, the simplified equation overestimates $P_{\mathrm{app}}^{}$ by ~50%. This deviation occurs because one of the assumptions in the derivation of this equation breaks down. Namely, that the solute concentration in the acceptor and donor compartments at equilibrium are the same.

The time evolution of the local concentration of solute in the aqueous compartments is shown in Figure 7A. The concentration of each solute species (protonated and deprotonated) in the outer compartment is equal to half the total solute concentration and does not change significantly over time due to the much larger volume of this compartment for the conditions considered ($c_{L}$ = 1 mM and $r_{o}$ = 100 nm). As expected, the concentration of the protonated species in the acceptor aqueous compartment at equilibrium is the same as that in the donor compartment. However, the concentration of deprotonated species equilibrates at a lower local concentration due to the decrease in this compartment’s pH (Figure 7B, black symbols).

In the simulations discussed above, it was assumed that the donor compartments had infinite buffer capacity, while pH variations in the acceptor compartments were only due to the permeation of the solute. However, the characterization of solute permeation by the pH variation assay requires the presence of a pH-sensitive probe, which behaves as a pH buffer. This buffer in the acceptor compartment attenuates the pH variation caused by the ionisation of the inflowing solute. Figure 7B illustrates this effect for a pH probe with a $pK_{a}$ = 7. As expected, the pH variation in the acceptor compartment decreases with increasing probe concentration and concomitant buffer capacity. For a local 100 µM probe concentration, $\Delta {\mathrm{pH}}_{i}$ is very small (<0.01).

The effect of increasing the buffer capacity in the acceptor compartment is further explored in Figure 8 for the case of vesicles with $r_{o}$ = 100 nm. As the concentration of pH-sensitive probe is increased, the equilibrium concentration of solute in the acceptor compartment increases, being essentially equal to that in the donor compartment for $P_{T}$ = 100 µM (Plot A). The equilibration characteristic constant decreases as more solute needs to permeate to reach equilibrium. It approaches half the value observed for the non-ionisable solute, as expected (Plot B). The value calculated for $P_{\mathrm{app}}^{r}$ is strongly dependent on $P_{T}$, reflecting the variation observed for β. However, $P_{\mathrm{app}}$ is essentially independent on $P_{T}$ because the decrease observed in β is compensated by the increase in $S_{i}\left(\infty \right)$. At the highest buffer capacity considered, the generated $\Delta pH_{i}$ is very small and the system essentially achieves full equalisation between internal and external solute concentrations. Under these conditions, the permeability coefficient calculated from both approaches differs by less than 10%. However, the generation of a significant $\Delta pH_{i}$ is required to allow following the solute dynamics from the variations in the signal from the pH-sensitive probe.

Figure 9 shows the results for the permeation of weak acids with different ionisation constants ($pK_{a}$ from 4 to 10) for a moderate buffer capacity inside the vesicles ($P_{T}$ = 10 µM). As expected, an increase in $pK_{a}$ leads to a faster equilibration of solute because the fraction of the neutral acid form increases (Plot A). The limit value attained for $pK_{a}\gg pH_{\mathrm{wo}}$ (*β* = 0.06 s^−1^) is the same observed for non-ionisable solutes permeating into vesicles with the same size (Figure 2). However, normalizing *β* by the fraction of neutral solute in the donor compartments ($\beta /fSH_{o}$) does not eliminate the dependence with the solute’s $pK_{a}$, the normalized value being larger for low $pK_{a}$ values. The fraction of solute in the acceptor compartments at equilibrium also depends on the solute’s $pK_{a}$. The variations of $\beta /fSH_{o}$ and $nS_{i}\left(\infty \right)/nS_{T}$ are related to the deviations from equalisation of the solute concentrations between donor and acceptor compartments, due to the different magnitude of $\Delta pH_{i}$ generated (Plot B). The $\Delta pH_{i}$ decreases as the solute $pK_{a}$ increases due to the higher contribution from the neutral acid species, becoming zero when $pK_{a}\gg pH_{\mathrm{wo}}$. The calculated permeability coefficient is shown in Plot C, for both $P_{\mathrm{app}}^{}$ and $P_{\mathrm{app}}^{r}$. A strong dependence with the solute’s $pK_{a}$ is observed as expected, with both estimates approaching that for non-ionising solutes with the same intrinsic parameters as the permeating *SH* species. However, while for $P_{\mathrm{app}}^{}$ this is essentially due to the variations in the fraction of *SH*, and $P_{\mathrm{app}}^{}/fSH_{o}$ becomes almost independent on the solute’s $pK_{a}$, for $P_{\mathrm{app}}^{r}/fSH_{o}$ a significant dependence is observed. This shows that $P_{\mathrm{app}}^{r}$ overestimates the permeability of solutes when the fraction of their charged species is significant. The % deviation between $P_{\mathrm{app}}^{r}$ and $P_{\mathrm{app}}^{}$ is shown in Plot B, being 10% when $pK_{a}=pH_{\mathrm{wo}}$, and over 30 % for $pK_{a}$ more than one unit lower than $pH_{\mathrm{wo}}$. Those deviations are a consequence of the $\Delta pH_{i}$ generated leading to non-equalisation of the solute concentration in the donor and acceptor compartments at equilibrium, and break-down of the assumptions in the derivation of $P_{\mathrm{app}}^{r}$.

#### 3.2.2. Lipophilic Solutes

For lipophilic weak acids or bases, the neutral species usually associates more efficiently with the lipid membranes [7,24,41,50,51,52,53]. In this section, we consider that the charged species has a 10-fold lower membrane affinity ($K_{P}^{\mathrm{SD}}=0.1\;K_{P}^{\mathrm{SH}}$). Given the thermodynamic cycle involving the membrane partition of both solute forms and the ionisation equilibria in the two compartments (aqueous and membrane), microreversibility constraints imply that the ionisation constant in the membrane must also be 10-fold lower than that in the aqueous medium ($pK_{a}^{\mathrm{Sl}}=pK_{a}^{\mathrm{Sw}}+1$). In order to evaluate the effect of increasing solute lipophilicity, $K_{P}^{\mathrm{SH}}$ was varied from 10^2^ to 10^4^.

The results obtained for a solute with $pK_{a}^{\mathrm{Sw}}=7$ are shown in Figure 10. As observed for non-ionisable solutes, the rate of equilibration increases sub-linearly with $K_{P}^{\mathrm{SH}}$, levelling-off for high $K_{P}^{\mathrm{SH}}$ values, where all the solute associates with the membrane (note the logarithmic scale in the x axis). For the conditions considered in the simulations $(pH_{\mathrm{wo}}=pK_{a}^{\mathrm{Sw}}=7)$, half of the solute in the outer aqueous phase is protonated. However, because the partition coefficient of the deprotonated species is 10-fold lower than that of the protonated one, the solute behaves as a weaker acid when associated with the membrane, and $fSH_{\mathrm{lo}}$ is equal to 0.91 for the solute in the membrane. For this reason, the equilibration rate for the non-lipophilic weak acid is about half of that of a non-ionising non-lipophilic solute, but their ratio approaches 0.91 as the weak acid’s lipophilicity increases (Figure 10A).

As observed for the case of non-ionisable solutes, the increase in solute lipophilicity also caused an increase in the amount of solute transported because the volume ratio of lipidic-to-aqueous phase is larger in the inner vesicle compartments. However, in the case of weak acids, this effect is counterbalanced by the pH variation, which hinders the full equalisation of the solute concentrations and is more extensive for solutes with higher lipophilicity (Figure 10B).

The calculated permeability coefficients are shown in Figure 10C. The simplified equation ($P_{\mathrm{app}}^{r}$) underestimates the permeability coefficient of very lipophilic solutes by orders of magnitude, being 75% smaller than $P_{\mathrm{app}}^{}$ for solutes with a moderate lipophilicity ($K_{P}^{\mathrm{SH}}=10^{2}$). The inadequacy of $P_{\mathrm{app}}^{r}$ in describing solute dynamics is even more evident when analysing its dependence on the vesicles’ radius (Appendix A). While the value of $P_{\mathrm{app}}^{}$ is independent of vesicle size, $P_{\mathrm{app}}^{r}$ decreases as the radius of the liposomes increases. The inaccuracy of the simplified equation increases with the lipophilicity of the solute, due to the larger pH variation inside the vesicles (Appendix A) and, thus, to stronger solute concentration gradients at equilibrium.

### 3.3. Following the Solute in the Aqueous Compartment Only

In the sections above, all solute that permeated the barrier (the non-polar centre of the membrane) was included in the calculation of the permeability coefficient from Equation (4). However, in some experimental setups, only the solute in the aqueous media is considered. In addition, it may be argued that only the solute that has equilibrated from the membrane into the aqueous media of the acceptor compartments has effectively permeated. The additional solute that permeates influences the equilibration rate constant but remains sequestered in the membrane barrier. The consideration of only the solute in the acceptor aqueous compartment does not influence the value of $P_{\mathrm{app}}^{r}$ because this variable is not included in the equation used for the calculation. However, the general equation must be modified to include only the solute in the aqueous compartments,
(21)$$P_{\mathrm{app}}^{w}=\beta \frac{nS_{\mathrm{wA}}\left(\infty \right)\;}{nS_{\mathrm{wD}}\left(0\right)}\frac{V_{\mathrm{wD}}}{A}$$

The fraction of solute in the aqueous media of both donor and acceptor compartments depends on the partition coefficient and on the fractional volume of aqueous media and membrane ($fV_{\mathrm{wA}}=V_{\mathrm{wA}}/V_{T}$ and $fV_{\mathrm{lA}}=V_{\mathrm{lA}}/V_{T}$, and corresponding equations for the donor compartments):(22)$$nS_{\mathrm{wA}}\left(\infty \right)=nS_{A}\left(\infty \right)\frac{fV_{\mathrm{wA}}}{fV_{\mathrm{wA}}+K_{P}\;fV_{\mathrm{lA}}}\;;\;nS_{\mathrm{wD}}\left(0\right)=nS_{D}\left(0\right)\frac{fV_{\mathrm{wD}}}{fV_{\mathrm{wD}}+K_{P}\;fV_{\mathrm{lD}}}$$

Thus, by substituting Equation (22) into Equation (21), one obtains the relationship between the permeability coefficient calculated considering only the solute in the aqueous phases and that considering the total solute in the donor and acceptor compartments:(23)$$P_{\mathrm{app}}^{w}=P_{\mathrm{app}}\;fV_{\mathrm{wD}}\frac{1+K_{P}\;\frac{fV_{\mathrm{lD}}}{fV_{\mathrm{wD}}}}{1+K_{P}\frac{V_{\mathrm{lA}}}{V_{\mathrm{wA}}}}$$

For non-lipophilic solutes ($K_{P}\leq 1$), and if the volume of the lipid phase is much smaller than that of the aqueous phase, the two estimates of the permeability coefficient become identical. If the donor and acceptor compartments have the same volumes $(V_{\mathrm{wD}}=V_{\mathrm{wA}}$ and $V_{\mathrm{lD}}=V_{\mathrm{lA}}$), the relationship between $P_{\mathrm{app}}^{w}$ and $P_{\mathrm{app}}^{}$ is independent of the solute lipophilicity, but is affected by the relative volume of the aqueous and membrane compartments.

In most practical situations, the volumes of the donor and acceptor compartments are different, and the volume of the membrane may be significant, leading to different predictions of the permeability coefficient when following all the solute or only that in the aqueous compartments, Figure 11.

For non-lipophilic solutes (plot A) the two estimates deviate only for very small volumes of the acceptor compartment (small vesicle radius):$P_{\mathrm{app}}^{w}$ is 10 % lower than $P_{\mathrm{app}}^{}$ for vesicles with 200 nm radius, and deviates more significantly for smaller vesicles. The difference between the two increases drastically as the solute lipophilicity increases, due to the sequestration of solute in the membrane. For a moderate solute lipophilicity ($K_{P}$ = 100, plot C), $P_{\mathrm{app}}^{w}$ is less than 50 % of $P_{\mathrm{app}}^{}$ for vesicles smaller than 1 µm.

Figure 12 shows the effect of solute lipophilicity in more detail for vesicles with a 100 nm radius (these are the largest unilamelar vesicles that can be obtained by extrusion of MLVs; thus, they are commonly used in permeability experiments).

As the solute lipophilicity increases, $P_{\mathrm{app}}^{w}$ underestimates the solute dynamics and the calculated value of this permeability coefficient is much smaller than $P_{\mathrm{app}}^{}$ (Plots A and B). The reason for $P_{\mathrm{app}}^{w}$’s failure to describe the solute dynamics is that the equilibration characteristic constant (*β*) reflects the time required for the permeation of a much larger amount of solute than what is considered in the equation. That is, $\beta \;nS_{\mathrm{wA}}\left(\infty \right)$ is much smaller than the initial rate at which the solute crosses the barrier: ${\frac{dnS_{A}}{dt}|}_{t=0}$ as per Equation (2).

Figure 12 also shows that both $P_{\mathrm{app}}^{}$ and $P_{\mathrm{app}}^{w}$ level off for very lipophilic solutes. As discussed in Section 3.2.2, this is because in this case most of the solute in the donor compartment was already in the vicinity of the barrier (associated with the membrane leaflet facing the donor compartment). Further increases in solute’s lipophilicity do not accelerate the solute’s permeation because its local concentration in the vicinity of the barrier does not increase.

The permeability coefficient shown in grey was calculated from Equation (24) and depends linearly on solute lipophilicity. In this equation, the calculation of the driving force for solute permeation considers only the solute in the aqueous phase of the donor compartment as in $P_{\mathrm{app}}^{w}$. However, in order to be consistent with the equilibration characteristic constant, all the solute in the acceptor compartment is considered:(24)$$P_{\mathrm{app}}^{w*}=\beta \frac{nS_{A}\left(\infty \right)\;}{nS_{\mathrm{wD}}\left(0\right)}\frac{V_{\mathrm{wD}}}{A}$$

The dependence of the solute dynamics on its lipophilicity is shown in plot C. As the solute lipophilicity increases, the fraction of solute in the donor aqueous compartment at *t* = 0 tends to zero. Concomitantly, the fraction of solute in the membrane compartment increases and the system tends to half the solute in the donor and half in the acceptor compartments at equilibrium. The solute concentration in the donor and acceptor compartments are identical because the relative volume of donor and acceptor membrane is the same, despite the much larger total volume of the donor compartment. Because the amount of solute that permeates the barrier does not increase further, the equilibration rate constant levels off at its maximum possible value $\left(k_{\mathrm{loi}}^{S}+k_{\mathrm{lio}}^{S}\right)$.

It should be noted that the linear increase in $P_{\mathrm{app}}^{w*}$ with $K_{P}$ is an arithmetic result that does not reflect the dynamics of the system. The important observation is that increasing the solute lipophilicity above an intermediate value will not increase the characteristic constant for solute equilibration across the membrane (*β*). The lipophilicity threshold will depend on the volume of the lipid phase. When working with model membranes, lipid concentrations of 1 mM as considered in this model are commonly used, and the threshold occurs at moderate solute lipophilicity ($K_{P}$ ≥ 100). If one considers a cell monolayer in a common permeability assay in 12-well plates (A ≅ 10^−2^ dm^2^, *V*_D_ ≅ 10^−3^ dm^3^), the volume of the membrane phase in contact with the donor compartment is only slightly above 10^−5^% of the donor volume. In this case, significant deviations from the linear dependence of *β* with solute lipophilicity will occur only for $K_{P}$≥ 10^5^. However, in the in vivo situation of permeation through the endothelium of capillaries (a cylinder with a radius of ≅ 5 µm) [54] the fraction of the membrane volume in the donor compartment is much higher, and close to the conditions considered in the simulations shown in Figure 12. In this case, deviations from the predicted increase in the solute rate of permeation with its lipophilicity may again be significant for moderate solute lipophilicities.

### 3.4. Intrinsic Permeability Coefficient

The sections above have shown that the apparent permeability coefficient may be calculated using different equations, leading to distinct values and to a distinct dependence on the properties of the permeating solute and on the system geometry. What permeability coefficient best describes the ability of the solute to permeate membrane barriers? Moreover, is there an intrinsic permeability coefficient? How can it be calculated and how is it related to the distinct values calculated for the apparent permeability coefficient? This section will address these questions.

The intrinsic permeability coefficient (P) should only be dependent on the rate constant for crossing the barrier ($k_{\mathrm{loi}}^{S}$), and on the amount of permeable solute in the vicinity of the barrier ($nS_{\mathrm{lD}}$ for non-ionisable solutes). The latter is given by:(25)$$nS_{\mathrm{lD}}\left(0\right)=nS_{D}\left(0\right)\frac{K_{P}\;fV_{\mathrm{lD}}}{fV_{\mathrm{wD}}+K_{P}\;fV_{\mathrm{lD}}}$$

The initial rate of barrier crossing for non-ionisable solutes is thus,
(26)$${\frac{dnS_{A}}{dt}|}_{0}=k_{\mathrm{loi}}^{S}nS_{\mathrm{lD}}\left(0\right)=k_{\mathrm{loi}}^{S}nS_{D}\left(0\right)\frac{K_{P}\;fV_{\mathrm{lD}}}{fV_{\mathrm{wD}}+K_{P}\;fV_{\mathrm{lD}}}$$

When Equation (26) is used in the general equation for the calculation of the permeability coefficient, one obtains:(27)$$P_{\mathrm{obs}}^{}=k_{\mathrm{loi}}^{S}K_{P}\;h\;\frac{1}{fV_{\mathrm{wD}}+K_{P}\;fV_{\mathrm{lD}}}$$

If sequestration of solute in the membrane is negligible (equivalent to $fV_{\mathrm{wD}}\approx 1\gg K_{P}\;fV_{\mathrm{lD}}$), Equation (27) simplifies to Equation (28), and the permeability coefficient becomes independent of the system geometry.
(28)$$P_{}^{}=k_{\mathrm{loi}}^{S}K_{P}\;h\;$$

This is the intrinsic permeability coefficient. It is only dependent on the rate at which the solute crosses the membrane, and on the fractional amount of solute in the vicinity of the barrier per unit area (given by $K_{P}\;h\;=\frac{nS_{\mathrm{lD}}}{\left[S_{D}\right]}\;\frac{1}{A}$).

Membrane thickness, h, is the parameter that converts the volumetric concentration considered in the calculation of the partition coefficient $K_{P}$ into the surface concentration of solute in the vicinity of the barrier. If the whole membrane leaflet volume is used when characterising $K_{P}$, then it is the whole leaflet thickness that should be considered [17]. On the other hand, if the solute is located at a known well-defined depth in the membrane, the partition coefficient is best calculated with respect to the volume of this membrane region, and its thickness should be used when calculating the permeability coefficient [55].

The permeability coefficients calculated by Equations (24) and (28) are equivalent and both correspond to the intrinsic permeability, $P_{\mathrm{app}}^{w*}=P$. In a real system, if the volume of the membrane becomes significant, and/or if a significant amount of solute is sequestered by the membrane, the observed permeability coefficient ($P_{\mathrm{obs}}$) becomes lower than the intrinsic one, this parameter being equivalent to the apparent permeability coefficient calculated from the general equation, $P_{\mathrm{obs}}^{}=P_{\mathrm{app}}$.

The above equations were derived for non-ionisable solutes. In the case of weak acids, if only the neutral protonated species permeates the membrane, the amount of permeable solute in the vicinity of the barrier is given by Equation (29), and $P_{\mathrm{obs}}^{}$ is given by Equation (30).
(29)$$nSH_{\mathrm{lD}}\left(0\right)=nS_{D}\left(0\right)\frac{K_{P}^{\mathrm{SH}}\;fV_{\mathrm{lD}}}{\left(1+\frac{K_{a}^{\mathrm{Sw}}}{H^{+}}\right)fV_{\mathrm{wD}}+\left(K_{P}^{\mathrm{SH}}+\frac{K_{a}^{\mathrm{Sw}}}{H^{+}}K_{P}^{\mathrm{SD}}\right)\;fV_{\mathrm{lD}}}$$
(30)$$P_{\mathrm{obs}}=k_{\mathrm{loi}}^{\mathrm{SH}}K_{P}^{\mathrm{SH}}h\frac{\;1}{\left(1+\frac{K_{a}^{\mathrm{Sw}}}{H^{+}}\right)fV_{\mathrm{wD}}+\left(K_{P}^{\mathrm{SH}}+\frac{K_{a}^{\mathrm{Sw}}}{H^{+}}K_{P}^{\mathrm{SD}}\right)\;fV_{\mathrm{lD}}}$$

The observed permeability is now dependent not only on the relative volumes of the membrane and aqueous compartment and on the membrane affinity of the permeating species ($K_{P}^{\mathrm{SH}}$), but also on solute’s ionisation equilibrium in the aqueous phase ($K_{a}^{\mathrm{Sw}}$), on the solution pH, as well as on the membrane affinity of the non-permeating species $(K_{P}^{\mathrm{SD}}).$ If sequestration of the solute in the membrane compartment is negligible, the above equation simplifies to Equation (31), where the observed permeability coefficient is proportional to the fraction of solute in the permeating form ($fSH_{\mathrm{wD}}$) [36].
(31)$$P=fSH_{\mathrm{wD}}\;k_{\mathrm{loi}}^{\mathrm{SH}}K_{P}^{\mathrm{SH}}h\;$$

The above Equations (28) and (31), allow the calculation of the intrinsic permeability coefficient from the parameters for the interaction of the solute with the membrane (translocation rate constant and lipophilicity) and the solute’s ionisation equilibrium. However, the effectiveness with which the solute permeates membrane barriers in real systems depends on the solute’s eventual sequestration by the membrane. If the relative volume of the membrane in the donor compartment can be estimated, the observed permeability coefficient may also be calculated. The comparison between the permeation effectiveness of different solutes in a given system is preferably done on the basis of $P_{\mathrm{obs}}^{}$, and will thus depend on the properties of the system under evaluation.

## 4. Conclusions

In this work, we developed a kinetic model for the permeation of solutes between two aqueous compartments separated by a lipid membrane. The major difference relative to other models available in the literature is the consideration of the lipid membrane as two finite compartments: the two membrane leaflets in contact with the donor and acceptor aqueous media. The permeation barrier considered was therefore not the whole membrane, but rather the non-polar membrane centre between the two leaflets. This is in agreement with the results obtained for the energy profile of drug-like molecules across lipid membranes, which usually shows a minimum at each membrane leaflet and an energy barrier at the membrane centre [30,56,57,58,59,60,61,62].

The model was developed for the case of non-ionizable solutes and weak acids, but it is easily adapted for the case of weak bases. The results for the latter case are essentially equivalent, except for the faster permeation of the deprotonated species (here the neutral form) leading to an increase in the pH inside the vesicles. The equations required for the implementation of the model for weak bases are provided in the Appendix A. On the other hand, the model cannot be directly applied to permeation through heterogeneous membranes and/or when solute association significantly changes the membrane properties (such as its surface potential). For small monovalent solutes, the latter should not be a concern whenever the solute concentration is less than 5 % of the lipid concentration [17,63]. This model assumes fast equilibration of all solute species between the aqueous media and the adjacent membrane leaflet. This assumption may break down for solutes that interact very strongly with the membrane, in which case desorption from the membrane becomes the rate-limiting step in the overall permeation [6,33,34,35,36].

This kinetic model allowed for an assessment of various methods used to calculate permeability coefficients from solute dynamics and from the intrinsic parameters for the interaction of solutes with lipid membranes (affinity and translocation rate constant). Each method was evaluated on the basis of its ability to provide estimates of the permeability coefficient independent of the system geometry, and with the expected dependence on solute’s lipophilicity. The assessed methods included the definition equation for the permeability coefficient $P_{\mathrm{app}}^{}$ given by Equation (4), the simple equation most commonly used, $P_{\mathrm{app}}^{r}$ given by Equation (6), as well as corrections of the reference equations for (i) a non-negligible volume of the vesicles’ encapsulated aqueous media, $P_{\mathrm{app}}^{\mathrm{rV}}$, given by Equation (19), and for (ii) solute sequestration in the membrane when only the solute in the aqueous phase of the donor compartment is known, $P_{\mathrm{app}}^{w}$ given by Equation (23). The characterised systems included a non-ionisable solute and a weak acid where only the uncharged protonated species permeated the lipid membrane.

This systematic study yielded important conclusions as follows.

When following the influx of solutes into the lipid vesicles’ lumen, none of the methods yielded a size-independent permeability coefficient for very small vesicles ($r_{o}<50$ nm). This is because the high curvature breaks the symmetry of the membrane, Figure 3.

The simplified equation ($P_{\mathrm{app}}^{r}$) failed to describe the solute permeation ability in any of the following cases: (i) the lipid vesicles were too small or too large (Figure 3); (ii) there was significant association of the solute with the membrane, corresponding to a moderate or high solute lipophilicity (Figure 4, Figure 5 and Appendix A); or (iii) the solute did not fully equilibrate with the acceptor compartment due to the development of opposing driving forces. In this model, this arose from the $\Delta pH_{i}$ generated by the selective permeation of the uncharged species of weak acids or bases (Figure 6, Figure 7, Figure 8 and Figure 9 for non-lipophilic solutes and Figure 10 and Appendix A for lipophilic solutes). This limitation has been pointed out before, [31] though it had not been analysed in detail.

The results obtained show that to accurately estimate the solute’s permeability coefficient from its equilibration dynamics it is necessary to know the extent of association with the lipid membrane and the extent of equilibration between the donor and the acceptor compartments. When the solute’s dynamics is followed through the pH variation in the acceptor aqueous compartment, it is an experimental requirement that the system does not reach full equilibrium. Therefore, when using this method, it is necessary to estimate the amount of solute that permeates the membrane, which may be done if the $\Delta pH_{i}$ is quantified and the buffer capacity inside the lipid vesicles is known. This is of particular relevance when comparing the permeabilities of molecules with different ionisation equilibria, due to the dependence of the observed $\Delta pH_{i}$ on $pK_{a}-pH_{o}$.

Importantly, when only the solute in the donor and acceptor aqueous compartments is quantified, the permeability coefficient obtained ($P_{\mathrm{app}}^{W}$) is the intrinsic one, which is not affected by solute sequestration in the membrane. This intrinsic parameter reflects the solute’s intrinsic ability to permeate membrane barriers. It is directly proportional to the solute lipophilicity and to the rate of translocation through the barrier (assumed as the rate-limiting step), Equation (28), and is in agreement with the predictions by the Meyer–Overton relationship, Equation (20).

The intrinsic permeability coefficient is the relevant parameter for experiments where large aqueous compartments are separated by thin membrane barriers, as with black lipid membranes or cell monolayer assays. However, this permeability coefficient cannot be used to compare the permeation ability of distinct solutes in real systems. For this purpose, the sequestration of the solute by the membrane must be taken into consideration. This is because an increase in solute lipophilicity will not cause a proportional increase in the observed permeability coefficient when the fraction of solute sequestered in the membrane is already significant. Such saturation may be responsible for the saturation observed in the in vivo bioavailability of drugs as a function of the permeability coefficient measured through cell monolayers in the classical two-chamber permeability assays [2]. To estimate the permeability coefficient that will be observed in a real system from the intrinsic one, it is necessary to know the fraction of solute associated with the membrane. This fraction can be calculated from the relative volumes of the membrane and aqueous phase in the donor compartment and the solute’s water to membrane partition coefficient. Sequestration of the solute by binding agents in the aqueous donor compartment will also influence the observed solute permeability. This aspect is not the focus of the present work, as it has already been addressed before (e.g., [6,55]).

As a general conclusion, when considering the permeation of drug-like molecules between two aqueous compartments separated by lipid membranes, the membrane cannot be considered as an infinitely thin barrier. Instead, the volume of the membrane leaflet in each side of the barrier must be accounted for, leading to a four-compartments system. Quantitative analyses of the dynamics of drug-like molecules through lipid membranes should be based on these models. This will allow one to dissect the observed permeability coefficient into the contributions from the solute lipophilicity and those from the intrinsic rate of barrier crossing. Discrimination of these contributions will improve the ability to predict drugs’ availability in tissues protected by tight endothelia such as the brain. This is a much-needed breakthrough towards decreasing the attrition rate in the development of new drugs for pathologies of the central nervous system.

## Figures and Tables

**Figure 1 membranes-12-00254-f001:**
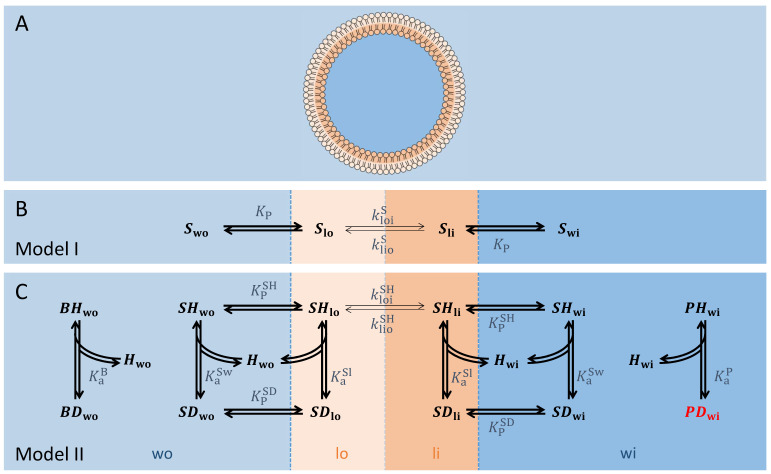
(**A**) Schematic representation of a liposome and the four distinct compartments. The outer aqueous compartment is represented in light blue, the outer membrane leaflet in light orange, the inner membrane leaflet in orange and the inner aqueous compartment in blue. (**B**) Reaction scheme for the permeation of a non-ionisable solute (Model I). The translocation between the membrane leaflets is the rate-limiting step (thin arrows), with the aqueous/membrane partition being at quasi-equilibrium (thick arrows). (**C**) Reaction scheme for the permeation of a weak acid (Model II). The translocation between the membrane leaflets is the single rate-limiting step (thin arrows), with protonation/deprotonation and aqueous/membrane partition processes being at quasi-equilibrium (thick arrows).

**Figure 2 membranes-12-00254-f002:**
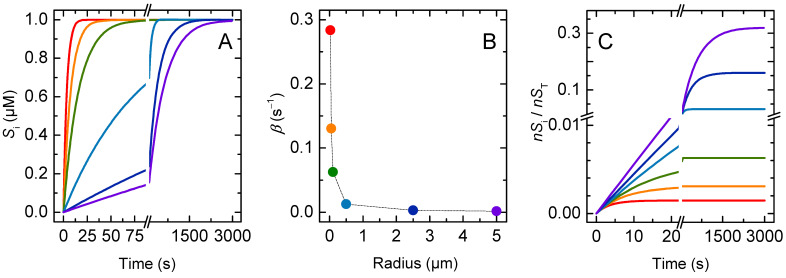
Effect of the vesicle size: (**A**) Dynamics of the total solute concentration in the inner compartments ($S_{i}^{}$), (**B**) Characteristic constant of solute accumulation inside the vesicles, and (**C**) Evolution of the fraction of solute in the inner compartments, for vesicles with a radius of 25 nm (**▬**), 50 nm (**▬**), 0.1 μm (**▬**), 0.5 μm (**▬**), 2.5 μm (**▬**) and 5 μm (**▬**).

**Figure 3 membranes-12-00254-f003:**
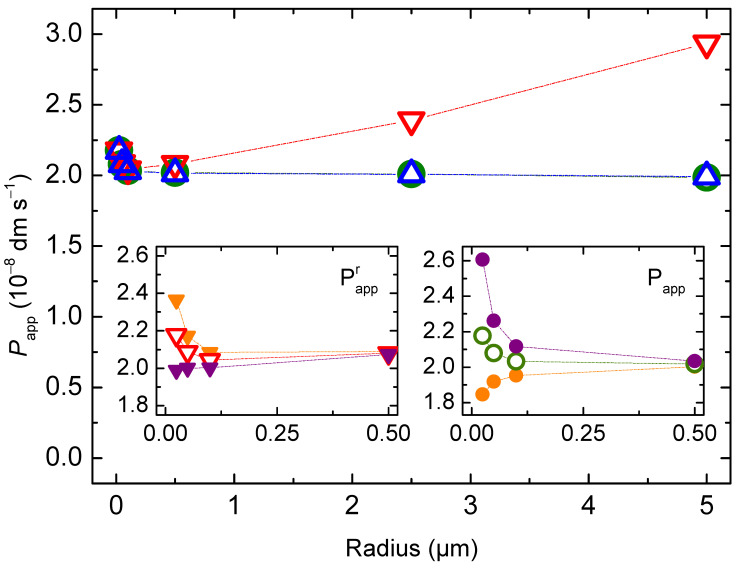
Apparent permeability coefficient calculated from the general equation ($P_{\mathrm{app}}^{}$ **○**), and from the simplified equation ($P_{\mathrm{app}}^{r}$ **▽** and $P_{\mathrm{app}}^{\mathrm{rV}}$
Δ), as a function of the vesicle’s outer radius, with the total area of the barrier calculated from the interleaflet radius of the vesicles ($r_{\mathrm{io}}$ ), Equation (12). The insets shows the dependence for vesicles smaller than 500 nm and considering the outer (● and ▼), the inner (● and ▼), or the interleaflet (**○** and **▽**) vesicles’ radius in the calculation of the total surface of the barrier; for $P_{\mathrm{app}}^{r}$ (left) and $P_{\mathrm{app}}^{}$ (right). The dotted lines are guides for the eye.

**Figure 4 membranes-12-00254-f004:**
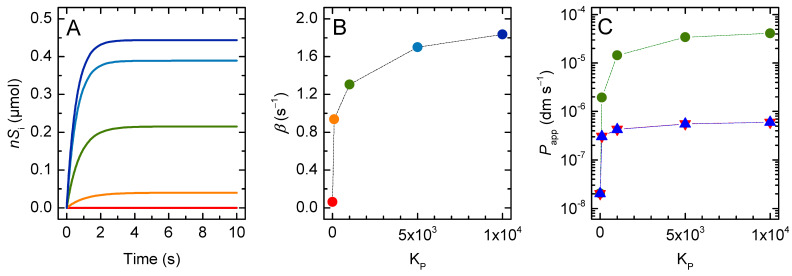
(**A**) Evolution of the number of moles of solute inside the vesicles when varying the partition coefficient of a non-ionisable solutes for partition coefficients 1 (**▬**), 10^2^ (**▬**), 10^3^ (**▬**), 5 × 10^3^ (**▬**) and 10^4^ (**▬**). (**B**) Characteristic constant of solute accumulation inside the vesicles as a function of the partition coefficient. (**C**) Apparent permeability coefficient calculated from the general equation ($P_{\mathrm{app}}^{}$ **●**), and from the simplified equation ($P_{\mathrm{app}}^{r}$ **▼** and $P_{\mathrm{app}}^{\mathrm{rV}}$
▲), as a function of the partition coefficient, with the total area of the barrier calculated from Equation (12). Note the logarithmic scale of the y axis. The dotted lines in panels B and C are guides for the eye.

**Figure 5 membranes-12-00254-f005:**
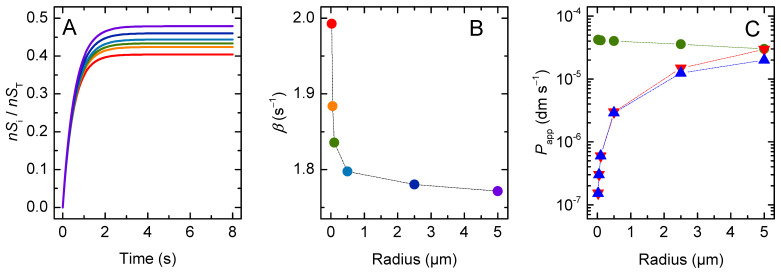
Permeation of very lipophilic solutes, *K_p_* = 10^4^. (**A**) Variation of the fraction of solute molecules inside the vesicles (membrane inner leaflet *plus* aqueous medium) with 25 nm (**▬**), 50 nm (**▬**), 0.1 μm (**▬**), 0.5 μm (**▬**), 2.5 μm (**▬**) and 5 μm radius (**▬**). (**B**) Effect of the vesicle radius on the characteristic constant of solute equilibration. (**C**) Apparent permeability coefficient calculated from the general equation ($P_{\mathrm{app}}^{}$
●), and from the simplified equation ($P_{\mathrm{app}}^{r}$
▼ and $P_{\mathrm{app}}^{\mathrm{rV}}$
▲). Note the logarithmic scale of the y axis. The dotted lines in panels B and C are guides for the eye.

**Figure 6 membranes-12-00254-f006:**
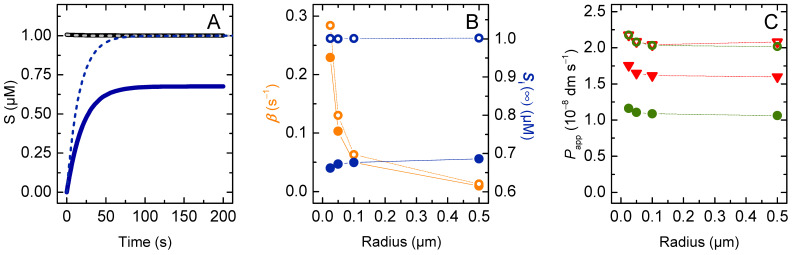
(**A**) Evolution of the local concentration of the total solute in the outer (**▬**, **▪ ▪ ▪**) and inner (**▬**, **▪ ▪ ▪**) aqueous compartments. The concentration of the corresponding species in the membrane is the same because $K_{P}$ = 1. (**B**) Variation of the equilibration characteristic constant (●, ○), and of the concentration of solute in the inner compartments at equilibrium (●, ○) as a function of the vesicle’s outer radius. (**C**) Apparent permeability coefficient ($P_{\mathrm{app}}^{}$
●, ○; and $P_{\mathrm{app}}^{r}$ **▼**, **▽**), as a function of the vesicle’s outer radius. The dashed lines and hollow symbols are the results for the permeation of a non-ionisable solute, and the solid lines and filled symbols are for the permeation of the protonated form of a weak acid with the same membrane affinity and flip-flop rate constant as the non-ionisable solute and $pK_{a}$ = 7. The lines in plots B and C are guides for the eye.

**Figure 7 membranes-12-00254-f007:**
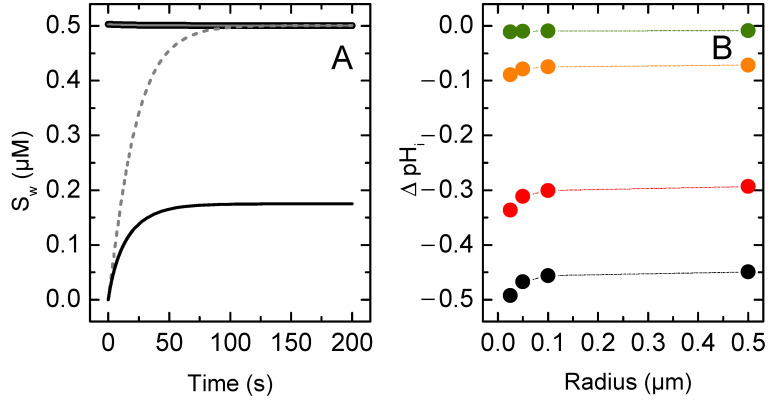
(**A**) Evolution of the concentration of solutes’ protonated species ($SH_{\mathrm{wo}}$**▬**, and $SH_{\mathrm{wi}}$ **▪ ▪ ▪**), and deprotonated species ($SD_{\mathrm{wo}}$ **▬**, and $SD_{\mathrm{wi}}$  **▪ ▪ ▪**), for vesicles with a 100 nm radius in the absence of additional pH buffers inside the vesicles. Only the concentration of the species in the aqueous phase is shown, the concentration of the corresponding species in the membrane is the same because $K_{P}$ = 1. (**B**) Maximum pH variation inside the vesicles, ${\mathrm{pH}}_{i}\left(\infty \right)-{\mathrm{pH}}_{o}\left(\infty \right)$, as a function of the vesicle’s outer radius for different internal pH buffer capacities, provided by entrapped pH probe at a total local concentration of 0 μM (●), 1 μM (●), 10 μM (●) and 100 μM (●). The dotted lines in plots B are guides for the eye.

**Figure 8 membranes-12-00254-f008:**
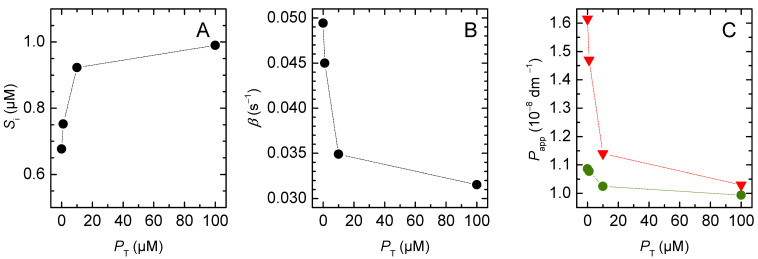
Concentration of solute in the acceptor compartments at equilibrium (**A**), and equilibration characteristic constant (**B**), as a function of the buffer capacity inside the vesicles. (**C**) Apparent permeability coefficient ($P_{\mathrm{app}}^{}$, circles, and $P_{\mathrm{app}}^{r}$,triangles) for the various buffer capacities, with the same colour code as in plots A and B. The dotted lines are guides for the eye.

**Figure 9 membranes-12-00254-f009:**
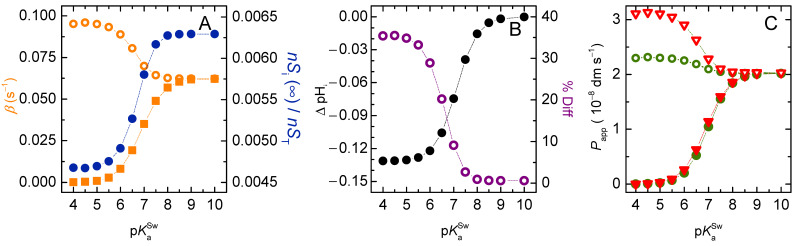
Effect of the solute’s acidity constant: (**A**) Variation of the characteristic constant (◼) and the ratio between *β* and the fraction of neutral solute in the outer compartments (**○**). Fraction of the concentration of solute in the inner compartments at equilibrium (●). (**B**) Maximum pH variation in the inner aqueous phase (●), and percentual error between the apparent permeability coefficients obtained from the general and the simplified equations (**○**). (**C**) Apparent permeability coefficient calculated from the general equation ($P_{\mathrm{app}}^{}$●), and from the simplified equation ($P_{\mathrm{app}}^{r}$ **▼**). The hollow symbols represent the ratio between the apparent permeability coefficient and the fraction of the protonated solute in the membrane. The dotted lines are guides for the eye.

**Figure 10 membranes-12-00254-f010:**
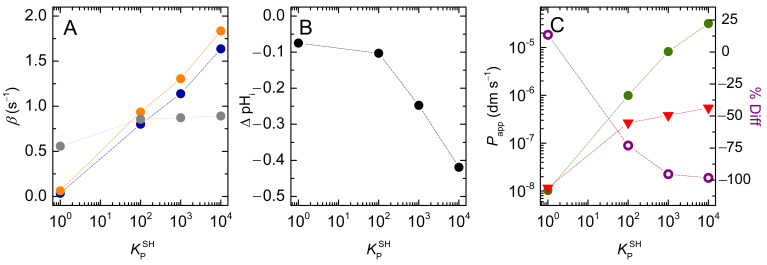
Effect of solute lipophilicity on: (**A**) the equilibration characteristic constant, for non-ionisable solutes (●) and weak acids (●), and ratio between the two (●); (**B**) the pH gradient generated at equilibrium by the permeation of the weak acids; and (**C**) the calculated permeability coefficient ($P_{\mathrm{app}}^{}$●,$P_{\mathrm{app}}^{r}$ **▼**), and % difference (100 × ($P_{\mathrm{app}}^{r}$ −$P_{\mathrm{app}}^{}$ )/$P_{\mathrm{app}}^{}$ **○**). Note the logarithmic scale of the x axis of all plots, and of the y axis in (**C**). The dotted lines are guides for the eye.

**Figure 11 membranes-12-00254-f011:**
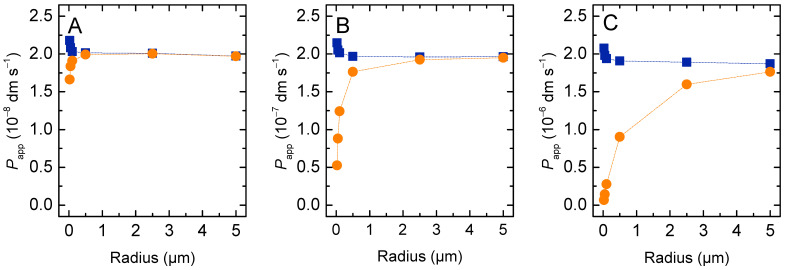
Effect of vesicle size and solute lipophilicity on the permeability coefficient calculated from the total amount of solute in the donor and acceptor compartments ($P_{\mathrm{app}}^{}$
■) and from the solute in the aqueous compartments only ($P_{\mathrm{app}}^{w}$
●), for $K_{P}$ = 1 (**A**), 10 (**B**) and 100 (**C**). The dotted lines are guides for the eye.

**Figure 12 membranes-12-00254-f012:**
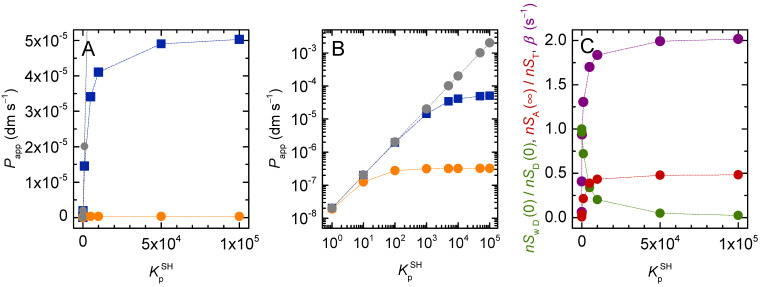
Calculated permeability coefficient as a function of solute lipophilicity ($P_{\mathrm{app}}^{}$■, $P_{\mathrm{app}}^{w}$
●, and $P_{\mathrm{app}}^{w*}$
●), in Cartesian (**A**) and log-log (**B**) scales. (**C**) Equilibration characteristic constant (*β*
●), fraction of solute in the aqueous compartment at *t* = 0 ($nS_{\mathrm{wD}}\left(0\right)/nS_{D}\left(0\right)$
●), and fraction of solute in the acceptor compartment at equilibrium ($nS_{A}\left(\infty \right)/nS_{T}$
●). The dotted lines are guides for the eye.

**Table 1 membranes-12-00254-t001:** Parameters used in the simulations of the Model I and II.

**Geometric** **parameters**	$r_{o}$	25 nm to 5 μm	**Partition** **coefficients**	$K_{P}$	1 to 10^4^
	*h*	3.96 to 3.94 nm		$K_{P}^{\mathrm{SH}}$	1 to 10^4^
	$a_{L}$	6.4 × 10^−17^ dm^2^ [38]		$K_{P}^{\mathrm{SD}}$	1 to 10^4^
	$V_{T}$	1 dm^3^	**Acidity** **constants (M)**	$K_{a}^{B}$	10^−7^
	$\bar{V_{L}}$	0.76 M^−1^ [39]		$K_{a}^{P}$	10^−7^
**Concentrations (M)**	$c_{L}$	10^−3^ M		$K_{a}^{\mathrm{Sw}},K_{a}^{\mathrm{Sl}}$	10^−7^ or 10^−4^ to 10^−10^
	$S_{T}$	10^−6^ M			
	$P_{T}$	0 to 10^−4^ M	**Rate constants** **(s^−1^)**	$k_{\mathrm{loi}}^{S}$	1 s^−1^
	$B_{T}$	0.01 M		$k_{\mathrm{lio}}^{S}$	$k_{\mathrm{loi}}^{S}\frac{V_{\mathrm{lo}}}{V_{\mathrm{li}}}$
	$H_{\mathrm{wo}}\left(0\right)$	10^−7^ M		$k_{\mathrm{loi}}^{\mathrm{SH}}$	1 s^−1^
	$H_{\mathrm{wi}}\left(0\right)$	10^−7^ M		$k_{\mathrm{lio}}^{\mathrm{SH}}$	$k_{\mathrm{loi}}^{\mathrm{SH}}\frac{V_{\mathrm{lo}}}{V_{\mathrm{li}}}$

$S_{T}$: Total concentration of the solute with respect to $V_{T}$; $P_{T}$: Total concentration of the fluorescent probe with respect to $V_{\mathrm{wi}}$; $B_{T}$: Total concentration of the buffer outside the liposomes, with respect to $V_{\mathrm{wo}}$.

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
