# Peer review of "Calculation of Permeability Coefficients from Solute Equilibration Dynamics: An Assessment of Various Methods"

_membranes, 2022, doi:10.3390/membranes12030254_

Round 1

Reviewer 1 Report

The manuscript “Calculation of Permeability Coefficients from Solute Equilibration Dynamics: An Assessment of Various Methods” by Margarida M. Cordeiro et al. has developed a new kinetic model of permeation that outperforms the old one, which is inherently applicable to very polar permeants only but has been inappropriately used in many more cases. The authors have shown the old model does not work for the permeation through very small or large vesicles, as well as the permeants with moderate or high lipophilicity. The authors have also proved the viability of their model in several specific cases. The merit of this work, based on my understanding, is that the membrane in permeation cannot be simply treated as an infinitely thin barrier and the permeant accumulation inside the membrane leaflets should be carefully accounted for. Overall, I find this paper is an excellent kinetic study of permeation with a well-structured and clear presentation. I believe this is a high-quality paper that merits publication in Membranes. Some minor questions for additional discussion are provided below. Hopefully the authors may find them useful.

1. The new kinetic model was developed for the permeation of weak acids. I think relevant findings and conclusions should also apply to weak bases. And I expect some changes to the corresponding equations (especially those in Appendix B-II). Please correct me if I am wrong. I am not requesting the authors to redo all their work for weak bases. But can the authors give some larger outlook in the Conclusions over the generalization of this study as implied to weak bases?

2. It reads to me that while deriving the model, the authors have not considered membrane composition, e.g., pure lipids or lipid mixture, if lipids are charge-neutral. These factors can substantially impact the permeation of ionizable molecules. Can the authors explain how considering membrane composition may or may not change the kinetic model in this work?

Author Response

Dear reviewer,

please find attached the answer to your comments.

We thank the reviewer for the carefull reading of the manuscript and for point out those questions which increased and clarified the scope of
the work.

Reviewer 2 Report

The article is interesting however the reviewer could not check the majority of the references most probably due to some issue with the reference manager software (lines 65, 67, 78,79, 83 are just some example). In general, the reviewer believe the manuscript is too difficult to follow as the amount of data presented is overwhelming. It is recommended to include in the manuscript only the most relevant details, and move all the other data in a supporting information section to which the authors can refer in case they need. 

Moreover, an abbreviation list section would be welcomed because there are many abbreviations and some of them are not even described (nSD, for instance).  

Line 119 onwards : this is anticipating the results and should be removed or moved to the results and discussion section.  

Author Response

Dear reviewer,

thanks for the carefull reading of the manuscript and for the helpfull suggestions. Please find the detailed answer to all your concerns in the attached file.

Round 2

Reviewer 2 Report

The article can be published in its present form.